# Study on the Applications and Regulatory Mechanisms of Grafting on Vegetables

**DOI:** 10.3390/plants12152822

**Published:** 2023-07-30

**Authors:** Wenjing Nie, Dan Wen

**Affiliations:** 1Huang-Huai-Hai Region Scientific Observation and Experimental Station of Vegetables, Ministry of Agriculture and Rural Affairs, Shandong Key Laboratory of Greenhouse Vegetable Biology, Shandong Branch of National Improvement Center for Vegetables, Institute of Vegetable Research, Shandong Academy of Agricultural Sciences, Jinan 250100, China; cottage1990@163.com; 2Yantai Key Laboratory for Evaluation and Utilization of Silkworm Functional Substances, Shandong Institute of Sericulture, Yantai 264001, China

**Keywords:** vegetable, grafting, mechanism, regulation, applications

## Abstract

Grafting can overcome problems with soil sensitivity, enhance plant stress tolerance, improve product quality, and increase crop yield and value. This paper reviews the various mechanisms of vegetable grafting, the graft survival process and its influencing factors, the practical applications of grafting, and the molecular regulation of grafting in vegetables. The importance of germplasm and rootstock interactions, the mechanization of vegetable grafting, and future aspects, including intelligence and digitalization, are discussed.

Grafting—the process whereby a branch or bud (termed a “scion”) from one plant is joined with the stem or root (commonly known as “rootstock”) of another organism within the same species—plays an instrumental role in artificial plant propagation. Cross-pollinated flora often pose challenges in the preservation and inheritance of beneficial traits from the maternal line due to interference from paternal DNA. However, grafting can effectively circumvent these issues. By purposefully amalgamating shoots or buds from superior specimens onto a compatible rootstock, it is possible to perpetuate the favorable genotypic traits of the maternal plant. This procedure results in the generation of economically beneficial clones that embody the preferred traits of the maternal genotype [1,2]. This fusion of distinct plant elements fosters the growth of an integrative plant, combining attributes from both progenitor organisms.

Grafting technology has garnered considerable attention in recent years, finding extensive application spanning horticulture, agriculture, and biology. It offers a versatile platform to investigate a broad spectrum of research areas, including floral initiation [3], bud proliferation [4], elucidation of heavy metal detoxification mechanisms, understanding of nutritional status [5], studies of soil-borne diseases [6], exploration of small RNA mobility, and examination of post-transcription silencing signals [7,8]. Its crucial role in elucidating the functioning of messenger RNA molecules [7] highlights its importance as a prime agricultural strategy. The technique serves to enhance crop yield, augment product quality, and augment crop resilience against various environmental stressors, leading to substantive economic acceleration. 

In this review, we will provide a detailed discourse on the multifarious processes and mechanisms incumbent in vegetable grafting. We also explore the factors influencing graft survival, the practical executions of the procedure, and deconstruct the molecular underpinnings fundamental to grafting dynamics within vegetable crops.

## 1. Common Methods of Grafting

Grafting methods in vegetables include cuttage grafting, cleft grafting, patch grafting, casing grafting, approach grafting, and double-root-cutting grafting [9,10] (Figure 1).

### 1.1. Cuttage Grafting Method (Figure 1A)

The cuttage grafting method entails the careful removal of the rootstock seedling’s true leaves and growth points using a surgical blade. To facilitate the grafting process, a bamboo stick of similar thickness as the scion is inserted into the base of the rootstock’s true leaves, leaving approximately 0.5 cm of the hypocotyl node. Simultaneously, the scion seedling’s hypocotyl is shaped into a wedge shape. The scion is then firmly inserted into the prepared area, negating the necessity for grafting clamps. This approach offers a straightforward and user-friendly procedure, resulting in a graft junction that is positioned above the soil surface, thereby reducing the risk of scion self-rooting.

### 1.2. Cleft Grafting Method (Figure 1B)

Rootstock and scion seedlings of similar stem thickness are cut across the hypocotyl, the lower part of the scion is cut into a wedge, and then the rootstock is cut into a cleft from the middle of the cross across the cut surface. A small opening is split from top to bottom, and then the scion is inserted into the cut of the rootstock to secure the graft.

### 1.3. Patch Grafting Method (Figure 1C)

This method requires cutting off the true leaf, cotyledons, and the growing point of the rootstock plant to form an oval incision about 1 mm long. The scion seedlings should be cut diagonally under the cotyledon, and the incision size should be the same as that of the rootstock incision. Then, the scion should be attached to the rootstock incision along the slope of the incision and fixed. The grafting method requires that the embryo axis of the stock and scion should be as close as possible to facilitate wound healing. This patch grafting method has the advantages of fast grafting speed, high survival rate, good interface healing, fast scion recovery and growth, and wide adaptability to a range of vegetables.

### 1.4. Casing Grafting Method (Figure 1D)

The rootstock and scion seeds are sown at the same time, and when the seedlings are at a suitable size for grafting, the rootstock hypocotyl is cut diagonally above the cotyledon and below the true leaf. The scion seedling is cut at the same position and at the same angle, and a plastic sleeve (-edge opening) is used to cover the rootstock hypocotyl, and then the scion section is inserted into the plastic sleeve, paying attention to ensuring that the two sections of the rootstock and the scion are closely aligned. As the grafted seedling grows, the plastic sleeve can be automatically removed. 

### 1.5. Approach Grafting Method (Figure 1E)

First, the scion and rootstock seedlings are cut at the same height, then the scion and the rootstock seedlings are lined up together, and the two cut surfaces are joined and fixed. After survival of the graft is confirmed, the upper part of the rootstock seedlings and the lower part of the scion seedlings (including its roots) are removed, and the remaining graft forms into a new grafted plant. This method is easy to operate and manage, with a high survival rate, because both the scion and the rootstock retain their roots in the early stage of grafting.

### 1.6. Double-Root-Cutting Grafting Method (Figure 1F)

As an emerging technology, double-root-cutting grafting has the advantages of a high survival rate of grafted seedlings, strong root vitality, vigorous growth, high grafting efficiency, short seedling rearing cycle, etc. In this grafting method, the scion is grafted onto the rootstock using a conventional grafting method, then the whole root system of the rootstock is removed from the base of the stem, and the resulting “cutting” is then cut again to improve the regeneration rate and vitality of the rootstock root system, which is essential for the generation of high-quality grafted seedlings. Exposure to 0.2 mmol·L^−1^ sodium hydrosulfide (NaHS) markedly promoted the growth and root system development of double-root-cutting-grafted tomato seedlings [11].

Root grafting is generally carried out when the rootstock seedling has grown one true leaf and the scion cotyledon is flat; the rootstock root is cut off with a blade, and the scion is cut to 4 cm long. The scion is attached to the rootstock by the cuttage grafting method. Immediately after grafting, the grafted seedlings are planted into the rooting substrate at a depth of about 1 cm, with the height of the graft above the ground at about 3 cm.

Due to the range of vegetables propagated by grafting, the methods of grafting are also diverse and not limited to the six methods listed, but these six methods are representative. In addition, the machine grafting method is being increasingly welcomed by the market. The uses of grafting are very broad due to the obvious advantages of grafted plants, but manual grafting of seedlings is laborious and time-consuming, especially for specialists in the seedling production base, where the cost is high. With the development of science and technology in recent years, automatic vegetable grafting machines have been developed that integrate mechanical and automatic control and horticultural technology. The machine can attach and fix the cut surfaces of a rootstock and a scion with a hypocotyl diameter of a few millimeters in a very short time, so that the grafting speed is greatly improved. At the same time, due to the rapid joining between the rootstock and the scion, long-term oxidation of the cut surface and loss of liquid from the tissues are avoided, which greatly improve the survival rate of grafting.

## 2. Process of and Factors Influencing the Successful Grafting of Seedlings

### 2.1. The Process of Successfully Grafting Seedlings

It takes 7~10 days for successful grafting of horticultural crop seedlings to be confirmed, with the process going through four stages. In the first stage, the incisions of the rootstock and scion are physically combined to form the contact layer, which is generally believed to be the deposit of some damaged cell walls and cell contents at the cut surfaces of the rootstock and scion [12,13]. This stage takes about 24 h. In the second stage, the cambium and parenchyma cells at the cut surface of the stock and scion divide strongly under the stimulation of the injury to form callus, which connects the rootstock and scion together. The cells between the rootstock and the scion begin to transfer and exchange water and nutrients through the plasmodesmata [12,13]. This stage takes 2~3 days. At the third stage, with the proliferation of callus between the rootstock and the scion, the contact layer disappears, and the callus between the rootstock and scion changes from being closely connected to being fused into one body, a change which can be difficult to identify. This stage takes 3~4 days. In the fourth stage, new vascular bundles develop in the callus at the rootstock/scion junction and connect with each other, and the rootstock and scion truly form a complete plant body [12,13]. In this way, water and mineral nutrients absorbed by rootstock roots can supply scion growth through the transport tissue, while photosynthetic products assimilated by the scion can be transported to the root through the transport tissue. This stage is generally completed 7~10 days after grafting.

### 2.2. Factors Influencing Successfully Grafted Seedlings

The survival rate of grafting is affected not only by internal factors, such as grafting affinity and rootstock and scion quality, but also by external factors, such as environmental conditions and grafting techniques.

#### 2.2.1. Internal Factors

##### Grafting Affinity

Grafting affinity refers to the ability of the rootstock and scion cambium to heal and survive and to grow together normally after close contact. Graft affinity is the most basic condition and the decisive factor for graft survival. The higher the degree of similarity between the rootstock and the scion in terms of shape, structure, physiological characteristics, genetic characteristics, and other aspects, i.e., the closer the kinship, the greater the grafting affinity and the higher the graft survival rate. For example, in vegetables, it is easier to graft between varieties of the same species, and it is more difficult to graft between species from different families.

##### Rootstock and Scion Quality

Rootstock and scion quality refer to the growth and development status and robustness of the rootstock and scion, which are factors directly affecting the graft survival rate. Well-developed, robust plants with high vigor and with no damage from diseases or insects store more nutrients and are more likely to survive after grafting, and their growth and development status after successful grafting is also higher. Diseased seedlings or weak seedlings are unlikely to survive after grafting; if they do survive, their subsequent development will be poor.

##### Rootstock and Scion Seedling Age

The age of the donor and recipient seedlings is also an important factor affecting the graft survival rate. Young seedlings exhibit slow healing and growth, whereas too-old seedlings exhibit high lignification of the stem, which is not conducive to the formation of callus. The appropriate seedling age of rootstock and scion partners should be determined according to the horticultural crop species and the grafting method [14,15].

#### 2.2.2. External Factors

##### Grafting Technology

Grafting technology is an important factor affecting the survival rate of grafts. In the grafting process, it is necessary to ensure that the cut surfaces of the rootstock and scion are smooth, the angle and length of the cut surface are appropriate, and that the stock and scion are closely connected. Concurrently, it is of utmost importance to maintain cleanliness during the grafting of the blades and secure them firmly. This promotes a stable, rather than a loose, connection—a crucial step to minimize unwarranted mechanical damage. A direct comparison between self-rooted, double-root-cutting grafting and single root-cutting grafting of watermelon seedlings suggests a more prosperous growth and developed root system in the double root-cutting grafting, with notable improvements in their phototrophic rates [16]. The investigation by Punithaveni et al. (2014) centered on the graft survivability of cucumber (*Cucumis sativus* L.) scions, in particular, the Green Long variety and the NS 408 hybrid. The grafting techniques scrutinized were side grafting (SG) and hole-insertion grafting (HIG), which were applied across a diverse range of rootstocks such as fig leaf gourd, pumpkin, winter squash, bottle gourd, and sponge gourd. The empirical findings accentuated a pronounced elevated success rate in grafting across all rootstock variations when employing the HIG method, in contrast to its SG counterpart [17].

##### Graft Management

After successful grafting, graft management is carried out to ensure that the scion does not wilt as a result of water loss; this is achieved by keeping the grafted seedlings under conditions of high humidity and low light. At the same time, to promote wound healing, the appropriate temperature needs to be maintained [13,18,19]. 

Temperature. Within a certain range, the higher the temperature, the faster the wound healing. For the three days after grafting, it is important to maintain an appropriately high temperature: for example, with cucumber, a daytime temperature of 25~28 °C and a night temperature of 17~20 °C are appropriate. For watermelon and melon, a suitable daytime temperature of 25~30 °C and a night-time temperature of 23 °C are appropriate, whereas for tomato, a daytime temperature of 23~28 °C and a night-time temperature of 18~20 °C are suitable. From 3 days after grafting, the temperature can be reduced by 1~2 °C. Excessively high temperatures should be lowered by appropriate shading and cooling. Generally speaking, the first 3 days after grafting is the key period for the survival of grafted seedlings [13,19].Humidity. High air humidity is conducive to the formation of callus, with dry air undermining the formation of callus by causing scion water loss. Before grafting, the rooting substrate should be soaked with water; immediately after grafting, spraying the grafted plants with water and covering them with a moisture-retentive film can be effective. Within the first 3 days after grafting, the air humidity should be maintained at 90~95%; from 3 d after grafting onward, the air humidity can be reduced to 85–90%, as appropriate [19,20].Light. For the process of graft survival, direct sunlight should be avoided as far as possible in order to reduce transpiration and to prevent the scion from wilting. In addition, low light is beneficial to callus growth. Therefore, after grafting, sunshade netting can be used to shade grafted plants from the sun; after 3~4 days, part of the sunshade netting can be removed to increase the amount of incident light, and 7~10 days after grafting, normal light management can be restored to the grafted seedlings [13,19,20].Gas. In order to meet the need for oxygen for respiration and carbon dioxide for photosynthesis of cambium cells at the junction of the rootstock and scion, the grafted seedlings also need to be properly ventilated, although attention needs to be paid to mist the seedlings after ventilation to prevent them from drying out [20].Pest and disease control. The occurrence of pests and diseases is also an important factor leading to grafting failure. Grafting operations can involve high-temperature and high-humidity conditions as well as major tissue damage caused by grafting, encouraging microbial infection and making the grafting contact area difficult to heal, resulting in plant disease and death [20]. Sucking insects, such as aphids and whiteflies, can be attracted to the graft junction and feed on young leaves of the scion, resulting in damage and possibly viral infection. A comprehensive prevention and control program against plant diseases and insect pests needs to be implemented in concert with a grafting program [13,19].

## 3. Influence of Grafting on Vegetables

### 3.1. Impact of Grafting on Abiotic Stress Tolerance in Crops

In plants, a wide range of abiotic stressors, such as salinity, drought, heavy metals, nutrient-deficient soil, and extreme temperatures can have deleterious impacts [21]. These negative effects may include ion toxicity, osmotic imbalance, metabolic dysfunction, and plasma membrane disruption. However, mitigation of these detrimental outcomes can be achieved by grafting. The use of rootstocks or scions with superior genotypes in grafting can alleviate the stressor-induced damage, ultimately promoting enhanced plant stress tolerance, growth, and development [22,23].

Grafting has been shown to improve a plant’s tolerance to salt, a stress that can adversely affect growth and development [24]. Effective regulation of osmotic substances, including proline, soluble carbohydrates, soluble proteins, and inorganic ions (K^+^, Na^+^, Cl^−^) by the rootstock allows plants to reduce their ion uptake and/or counteract the harmful impacts of salt [24]. This ultimately results in enhanced scion salt tolerance. Continuous glasshouse cultivation of cucumber was prevented because of its weak root system and sensitivity to damage from soil contaminated with salt and other harmful substances [1]. Replacement of the cucumber roots with black-seed pumpkin roots greatly reduced the damage caused by soil salts, etc.; despite the grafted cucumber plants having the same root area as the seedlings, the number of roots on the grafted plants was twice that on cucumber seedlings, resulting in more than 30% more nitrogen and potassium and about 80% more phosphorus being taken up by the grafted plants, which were able to access phosphorus reserves deep in the soil horizon [1]. Specifically, these rootstocks promoted the accumulation of osmoregulatory substances, such as proline and glycine betaine, as well as augmented the activities of antioxidant enzymes, including superoxide dismutase, peroxidase, catalase, and glutathione reductase. By enhancing a plant’s antioxidant enzyme activities, salt-induced damage as a result of oxidative stress can be effectively mitigated, as evidenced by the marked increase in such antioxidant enzyme activities in plants with salt-tolerant rootstocks. In environments with elevated salinity, grafted cucumber and tomato plants displayed adaptive mechanisms such as increased antioxidant enzyme activity [21,24]. In comparison with self-grafted plants, cucumber scion leaves grafted onto salt-tolerant pumpkin rootstocks exhibited heightened sensitivity to abscisic acid (ABA) under saline-stress conditions. This led to rapid stomatal closure, reduced transpiration, and decreased water loss, ultimately mitigating plant wilt and significantly enhancing the grafted cucumber’s capacity to cope with salt stress [25]. Under saline stress, the pumpkin rootstock triggered upregulation of the expression of key genes involved in ABA biosynthesis, such as *NCED2*, *ABCG22*, *PP2C*, and *SnRK2.1*, in the grafted cucumber plants. This heightened sensitivity of the rootstock facilitated ABA transfer from the roots to the scion, subsequently lowering the transpiration rate and stomatal conductance of the leaves and ultimately improving the grafted cucumber plant’s salt and drought tolerance [25,26].

Nevertheless, further investigation is required to ascertain the direct involvement of specific rootstocks and scions in achieving alterations in plant stress tolerance as well as the causal relationship between augmented microRNA (miRNA) expression and stress adaptation [27,28,29]. In response to drought stress, pumpkin rootstocks have been shown to enhance the drought tolerance of scions by modulating the expression of miRNAs [29]. However, it remains to be elucidated whether the variations in abiotic stress tolerance in pl ants can be directly attributed to the specific rootstocks and scions [27,28,29]. Furthermore, grafting has been demonstrated to bolster a plant’s resilience to waterlogging and suboptimal light conditions. In the context of flooding stress, watermelon leaves grafted onto rootstocks derived from pumpkin and zucchini exhibited a significant increase in stomatal conductance (*Gs*), transpiration rate (*Tr*), CO_2_ exchange rate, and dry matter mass compared with leaves on their self-rooted counterparts. Conversely, the chlorophyll content and reactive oxygen species (ROS) production rate were notably lower in self-rooted seedlings, and the antioxidant enzyme activities in self-rooted seedlings were higher than in the grafted plants, where the development of adventitious roots and aeration tissues, absent in the self-rooted seedlings, was observed [30]. Upon grafting, eggplant demonstrated enhanced nutrient uptake and endogenous hormone synthesis, increased water-use efficiency, and heightened tolerance to both low- and high-temperature stresses [31,32]. Moreover, grafted plants exhibited reduced absorption of harmful substances from the soil, increased tolerance to alkalinity, salinity, and flooding, as well as a decreased presence of boron [33,34] and copper [35]. Compared with nongrafted melon plants, the root of melon plants grafted onto the commercial Cucurbita maxima Duchesne x Cucurbita moschata Duchesne rootstock “TZ-148” had higher selectivity and lower boron absorption [34].

### 3.2. Impact of Grafting on Biotic Stress Resistance in Crops

During the grafting process, rootstocks and scions undergo various morphological, physiological, and biochemical changes at the point of union. Their genetic traits, tissue architecture, and defense mechanisms become interrelated. Consequently, grafted plants exhibit greater resistance to pests and diseases than self-grafted seedlings, particularly when grafted onto rootstocks of resistant varieties [36]. Extensive research has demonstrated that grafting is an environmentally sustainable and effective approach to mitigating soil-borne infections [37]. Significantly lower disease indexes for vine wilt were observed in melon plants grafted with pumpkin as the rootstock [38,39]. Cucumber grafted onto pumpkin can effectively improve nutrient uptake from the soil as well as prevent cucumber fusarium wilt and delay the occurrence of downy mildew at the same time [17,40]. Moreover, grafting eggplant decreased the incidence of yellow wilt disease, subsequently improving the plant’s ability to transport nutrients and synthesize endogenous hormones [31]. Numerous studies have demonstrated that grafting with disease-resistant rootstocks can enhance disease resistance in watermelon [30,41], melon [38,42,43], and other vegetables, particularly providing substantial improvements against soil-borne root diseases, such as fusarium wilt. Furthermore, grafting has been shown to boost resistance against foliar diseases such as blight, brown streak, powdery mildew, and viral infections while decreasing the prevalence of some diseases, such as cranberry blight [44,45].

Researchers like Robert (2009) and Warschefsky (2016) have proposed that grafted plants can transport hormones, mRNAs, small RNAs (sRNAs), and proteins over greater distances than in ungrafted plants, which may be the key to grafting’s ability to improve vegetables and to enhance their resistance toward pests and diseases [46,47].

### 3.3. Impact of Grafting on Crop Quality

Employing different grafting combinations elicits notable disparities in both the yield and quality of vegetable crops. Emphasizing the strategic optimization of these grafting methods is crucial in harnessing their inherent potential to boost the productivity and quality of these crops. The findings of studies focusing on changes in horticultural crop fruit quality following grafting remain a subject of debate. Empirical studies provide a robust foundation for this argument; the grafting of tomatoes onto various rootstocks led researchers to report that the dry matter, soluble solids, soluble proteins and titratable acid content values of fruits from the grafted plants were significantly higher than in self-rooted tomatoes [41,48]. The impact of different rootstock treatments on tomato fruit quality varies; however, it is evident that the quality of fruits from grafted seedlings surpasses that of self-rooted tomatoes [41,49]. Some researchers report enhanced fruit yield and quality in tomatoes as a result of specific grafting combinations that heighten the plant’s photosynthesis rate and nutrient absorption efficiency [1,11,41]. By grafting three watermelon cultivars onto each of three hybrid pumpkin rootstocks, Cushman et al. (2008) conducted a comparative study of grafted and ungrafted watermelon plants under commercial conditions in Florida. They employed a range of scions, namely, ‘Tri-X 313’, ‘Palomar’, ‘Petite Perfection’, and ‘Precious Petite’, and rootstocks, including none, BN111, BN911, ‘Emphasis’, J008, and ‘Ojakkyo’, in a randomized block design. The findings suggest that grafting led to a moderate increase in maturation time and an enhancement in plant productivity, while leaving fruit color and hollowheart ratings largely unaffected. Furthermore, it appeared to yield larger and firmer fruits. However, distinct scion/rootstock combinations produced variable total soluble solids content (TSS) [50]. There are also studies showing that cantaloupe cultivars were compatible with certain interspecific hybrid squash rootstocks but incompatible with others. Grafting tended to delay harvest but did not compromise fruit quality, yield was increased for some cultivars, and the highest fruit yield and number per plant were observed in specific grafting combinations [42].

Numerous studies have demonstrated that grafting onto various rootstocks can alter the type and concentration of carotenoids in melon fruits [39,42]. Furthermore, after grafting, changes in volatile compounds were observed in tomato fruits, resulting in a quality inferior to that of self-rooted tomatoes [48]. Under both glasshouse and open-field cultivation conditions, grafting was found to reduce the vitamin C content of tomatoes [49]. Factors such as different rootstock combinations, rootstock/scion affinity, delayed fruit development after grafting, and early harvesting may all contribute to the reduced flavor quality of grafted fruits [49]. Quality attributes of fruits harvested from determinate “Florida 47” tomato plants grafted onto either “Beaufort” or “Multifort” rootstocks were scrutinized against those without any grafting or those grafted onto themselves; the grafting process substantially elevated fruit yields when they were related to their non-grafted or self-grafted counterparts [49]. Furthermore, the yield of commercially viable fruit experienced a conspicuous augmentation, registering an increase of approximately 41% [49]. Habran et al. (2016) reported that suitable rootstocks can affect the activities of enzyme activities associated with phenolic biosynthesis and the concentration of phenolics, particularly flavonoid compounds such as anthocyanins, proanthocyanidins (condensed tannins), flavan-3-ols, and flavonols in scion fruit [51]. These effects lead to an enhancement in the fruit quality of the scion variety. Rootstock grafting has been shown to influence the color of watermelon flesh by altering its lycopene content; research by Soteriou et al. (2014) discovered that grafting onto the inter-specific *C. maxima* × *C. moschata* hybrid rootstock TZ14 led to a delay in changes to watermelon flesh color compared with self-rooted seedlings [14]. Meanwhile, Cushman et al. (2008) found that the accelerated development of watermelon female flowers due to rootstock grafting resulted in changes to fruit skin color; however, this was accompanied by a decline in both the soluble solids content and overall flavor quality [50].

The capacity of rootstocks to take up and assimilate nutrients, water, and other resources varies, which, in turn, alters the balance between vegetative and reproductive growth in the scion. Consequently, this influences the fruit phenotype and the accumulation of secondary metabolites such as flavonoids [51]. Significant differences exist among rootstock varieties, and these disparities yield varying effects. The interchange of resources between rootstocks and scions may also lead to alterations in the enzyme activities regulating flavonoid metabolism in the scion, or even cause changes in the expression of related genes. Additionally, rootstocks themselves exhibit differences in root secretions, hormone levels, and associated root microbiomes that further impact the growth and developmental processes of the scion of the grafted plant [52]. These factors could potentially affect the expression of genes linked to flavonoid metabolism, modify the activity of enzymes controlling flavonoid biosynthesis in the scion, and ultimately result in variations in flavonoid metabolism within the fruit. Numerous studies have revealed that employing pumpkin as the rootstock for grafted melon adversely affects the fruit’s flavor and aroma [39,53,54]. Melon rootstocks, owing to their enhanced affinity for melon scions, exhibit superior grafting compatibility and minimizes this effect on fruit quality post-grafting [42,55]. Furthermore, research has demonstrated a significant increase in lycopene and trace element concentrations in watermelon fruits when utilizing gourd, pumpkin, or wild watermelon as rootstocks [14,56,57].

In recent studies, the grafting of watermelon onto rootstocks derived from gourd, pumpkin, or wild watermelon has exhibited promising results. Notably, these grafting combinations have led to a substantial increase in lycopene and trace element concentrations in watermelon fruits. This enhancement in fruit quality can be attributed to the improved compatibility between the melon scion and the selected rootstocks. Furthermore, this grafting method minimizes any potential negative impacts on fruit quality that may arise from the grafting process itself.

Rootstock grafting influences the plant’s photosynthetic function and water absorption capacity, which, in turn, affects the fruit’s texture [50]. Grafting onto pumpkin rootstocks significantly increased the flesh firmness of watermelon fruits [58,59], whereas various cucurbit rootstocks showed different impacts on the hardness of watermelon fruits from grafted plants [58,60]. After grafting, the occurrence of umbilical rot in tomatoes decreased, revealing a strong correlation between rootstock/scion affinity and disease incidence [61]. Yetisir et al. (2003) and Alan et al. (2007) discovered that grafting watermelon onto pumpkin, cucurbit, or wild watermelon as rootstocks did not significantly alter the fruit shape or the fruit shape index [58,62]. Furthermore, it has been postulated that rootstock grafting influences the morphology of melon fruits, particularly watermelons [63]. Although Alan et al. (2007) observed no noticeable difference in rind thickness between grafted and self-rooted watermelons [62], it has been suggested that watermelon rind thickness is mainly determined by the scion variety [64].

In recent years, an increasing number of studies have been conducted to elucidate the grafting process at the molecular level to identify how grafting influences fruit quality. These investigations contribute to a deeper understanding of how rootstock/scion combinations can affect the texture, flavor, and overall quality of various vegetables. Garcia-Lozano et al. conducted a study examining transcriptome variations in watermelon scions grafted onto cucurbit rootstocks, revealing differential expression patterns of numerous genes involved in ripening and quality in the tissues of both grafting combinations [65]. In an investigation of the pericarp transcriptome of grafted cucumbers, Zhao et al. found that different pumpkin rootstocks could substantially alter gene expression related to sugar and aromatic compound synthesis, impacting fruit quality [66].

### 3.4. Impact of Grafting on Crop Yield

Grafting exhibits a significant potential for enhancing plant resistance and augmenting the root absorption capacity of rootstocks. Consequently, grafted seedlings exhibit rapid and vigorous growth, thereby stimulating higher crop yields [67]. Grafting enhances nutrient uptake, stimulates growth and development, and consequently increases yield [68] by controlling the metabolism of endogenous hormones such as cytokinins (CTKs), gibberellins (GAs), auxins (IAAs) and other growth regulators, primarily through the rootstock.

Lopez-Pérez et al. (2006) found that tomato cultivars grafted onto nematode-resistant tomato rootstocks exhibited higher yields than their ungrafted counterparts [69]. Rouphael et al. (2008) determined that the enhanced nutrient availability, water uptake and CO_2_ assimilation, mediated by the rootstocks contributed to the increased fruit production and water-use efficiency observed in grafted watermelon plants and to a notable improvement in the plant’s overall nutritional status [70]. In comparison with the autografting of “Jiaxina 74–112” with no salt stress, the utilization of “Western tomato rootstock” in grafted tomato plants resulted in a significant reduction in yield of 32.3% under salt stress conditions. Conversely, when subjected to the same salt stress, grafted tomatoes using “Western tomato rootstock” exhibited a remarkable 37.7% increase in yield [71].

Numerous studies have demonstrated that using a rootstock such as pumpkin for grafting can promote melon fruit enlargement and increase melon production [39,54,72,73]. A separate investigation revealed that the quality of individual fruits from grafted watermelon plants with a strong affinity between scion and rootstocks increased by 55%; in contrast, those combinations with low affinity exhibited diminished fruit quality and yield [74,75]. This evidence supports the concept that using grafted plants in greenhouse watermelon cultivation can substantially improve the individual fruit quality. To enhance fruit quality, it is imperative to select the appropriate rootstocks for grafting. The selection of rootstocks capable of enhancing the number of fruits per plant is a vital factor in forecasting high yields in grafted plants. It is important to note that overall yield is determined primarily by the number of fruits per plant rather than by the size of individual fruits. In cases where the fruit yield is limited, the scion’s genotype can be utilized to increase total yield by promoting a greater number of fruits per plant [50].

According to Schwartz et al. (2013), tomato plants grafted onto rootstocks exhibiting strong vigor exhibited increased mass per fruit, a greater number of fruits per plant, and significantly higher yields compared with those grafted onto less vigorous rootstocks [61]. In conclusion, the optimal rootstock/scion combination is crucial for attaining high yields, as it influences the growth, development, and yield of vegetables.

## 4. Molecular Regulatory Mechanism in Grafting

### 4.1. The Roles of Hormones and Metabolites in Grafting

Plant growth regulators, or plant hormones, are another crucial component influencing growth and developmental processes in plants. These endogenous biochemical substances are synthesized by the plant itself. During the grafting procedure, plant hormones play a dynamic role in the rootstock-scion interaction, particularly in callus formation. Extensive research supports the involvement of cytokinins, auxins, abscisic acid, gibberellins, jasmonic acid, and ethylene. Compounds such as the auxin indole-3-acetic acid (IAA) and the cytokinin 6-benzyladenine (6-BA) are notable for augmenting xylem and phloem transport efficacy [76]. In addition, IAA, CKs, JA, and GAs have a coordinating effect at the rootstock/scion contact area, which can stimulate cell differentiation and promote the formation of callus. IAA synthesized in the leaves of the scions can be transferred downward to the rootstock and promote the growth and development of rootstock lateral roots [77]. A study on the grafting system of pepper showed that by balancing the concentrations of CKs and ABA, the vigor of the scion was increased, thus improving the vigor of grafted plants [78]. CKs and GAs were transported to scions through the xylem and promoted branch growth and internode elongation of the scion partner [79]. Studies have shown that abscisic acid (ABA) is an important hormone regulating stomatal closure in the leaves of plants. When plants are under drought stress, this can induce the transport of ABA from the rootstock to the scion, regulating stomatal closure in leaves [80] and thus improving the drought tolerance of the grafted plants [81]. In addition, JA is an oxidized lipid that controls the expression of defense genes in plants in response to cell damage. Studies have shown that under osmotic stress conditions, scion plant leaves can be induced to synthesize a larger amount of JA and transport it to rootstock roots to alleviate the damage caused by abiotic stress in the plants [82,83]. Melatonin participated in the JA-enhanced cold tolerance of grafted watermelon plants [84]. In conclusion, hormones interact with each other in the scion and rootstock of grafted plants.

A substance consisting of β-1,4-glucanases secreted into the extracellular region facilitates cell wall reconstruction near the graft interface, and overexpression of the β-1,4-glucanase gene promotes grafting in the process of cell–cell adhesion [85]. Sugar metabolism is active and essential during graft union formation [86].

### 4.2. Unveiling Grafting Mechanisms through Genomic Analysis

Numerous substances in grafted plants possess the ability to traverse from the donor tissue to the accepting tissue via the phloem. These substances perform pivotal roles within the recipient organ, demonstrating their significant influence by regulating the transportation of minute molecules such as plant hormones and metabolites. Additionally, these substances also control the movement of macro molecules such as mRNAs, non-coding RNAs (ncRNAs), and small RNAs (sRNAs), which act as signal bearers in both intercellular and systemic signaling networks [87,88]. mRNAs are important genetic information materials. A large number of mRNAs can move between the rootstock and the scion of grafted plant systems. The mRNAs that can be transported for long distances are collectively referred to as mobile mRNAs (mob-mRNAs). It has been reported in the literature that the movement of mRNAs in plants is selective [89,90,91], as described in Arabidopsis/Benni tobacco, tomato/tobacco, tomato/potato, watermelon/pumpkin, watermelon/cucumber and many other grafting combinations [92].

Studies have shown that mob-mRNAs have an important effect on the development of recipient tissues, and some mRNAs encoding non-cellular autonomous proteins can regulate the growth and development process of the recipient tissues after translation, including their physiological growth, fruit development, and self-resistance [93,94]. Haywood et al. (2005) first reported the movement of *Gibberellic Acid-Insensitive* (*GAI*) mRNAs in pumpkin, followed by the movement of homologous transcripts in *Arabidopsis*, tomato, dodder, and apple [93,95,96].

With the development of sequencing technology, the mechanism of long-distance transport of mob-mRNAs is being elucidated. A large number of mob-mRNAs in grafted plants has been identified, and the regulatory mechanism and role of mob-mRNAs in grafted plants are gradually becoming clear. A total of 309 mob-mRNAs were identified in the cucumber grafting system, which were mainly associated with photosynthesis, oxidative phosphorylation, photosynthetic proteins, and ubiquitin-mediated proteolysis [81], whereas more than 3000 mob-mRNAs were found in watermelon/cucumber grafts [90]. In addition, 152 low-temperature-induced mob-mRNAs were identified in cucumber/pumpkin-grafted plants under low-temperature stress. Functional enrichment indicated that there was a relationship between cold-tolerant mob-mRNAs and the β-oxidative degradation of fatty acids in low-temperature-sensitive cucumber [97], while mob-mRNAs that move from scion to rootstock are also involved in pathways of carbon fixation and amino acid biosynthesis. A total of 111 mob-mRNAs were identified in a potato grafting system, and functional enrichment indicated that mob-mRNAs were involved in regulating pollen tube development and fruit morphology in potato [98].

In addition, studies have shown that mob-mRNAs can be transferred to target tissues and translated, thus participating in the regulation of the target tissues. For example, the tomato system protein precursor gene (PS) can be translated into PS protein in the scion partner [99]. Mob-mRNAs move through phloem tissues and play an important role in the regulation of processes in target tissues. The change in auxin concentration is considered to be an important event in vascular differentiation in the early stage of grafting [100], which may occur through the transcription factor MONOPTEROS (MP), also known as AUXIN RESPONSE FACTOR5 (ARF5). MP directly activates the transcription of the homeodomain–leucine zipper III gene, *ATHB8*, which is necessary for pre-procambial cell specification and the coordination of procambial cell identity.

### 4.3. The Role of Genes in Grafting

Current reports indicate that multiple genes and transcription factors play important roles in grafting. Many genes associated with callus formation and cell proliferation have been reported. *ANAC071* and *RAP2.6L* are two plant-specific transcription factor genes whose expression is promoted by ethylene and jasmonic acid, and together with the concomitant accumulation of indole-3-acetic acid, they are essential for tissue reunion in the graft [101]. Meanwhile, the expression of *ANAC071* is induced by auxin. *ANAC071* binds to the promoters of *XTH19* and *XTH20* and induces their expression and enhances cell proliferation in the tissue interaction process [102] (Figure 2). An asymmetrical gene regulatory mechanism has been reported. At the scion cut surface, *ANAC071* was expressed, with the concomitant accumulation of IAA [101]. In contrast, at the rootstock cut surface, *RAP2.6L* was expressed, with a concomitant decline in IAA concentration [101]. A recent report showed that *ANAC071* and *ANAC096* are redundantly involved in the process of “cambialization” [103].

Furthermore, Indole-3-acetic acid (IAA), pivotal in grafting, is modulated by PIN-formed proteins (PIN), AUXIN RESPONSE FACTOR 6 (ARF6), and ARF8. These factors control the auxin response factor, MONOPTEROS (MP), influencing the differentiation of xylem through Arabidopsis Histidine Phosphotransfer Protein 6 (AHP6) gene. Similarly, the differentiation of protophloem is regulated through the BREVIS RADIX (BRX) gene. Meanwhile, the ALF4 gene manages cell division and the xylem pole pericycle, whereas pre-procambial cell specification is governed by the Arabidopsis Thaliana homeobox 8 (ATHB8) gene. [100,104,105,106,107]. *ARR10* and *ARR12* are implicated in cytokinin-mediated regulation of protoxylem differentiation [108], whereas *WIND1* (*WOUND-INDUCED DEDIFFERENTIATION 1*) promotes callus formation through the cytokinin signaling pathway [109] (Figure 2).

*AOS* and *JAZ10*, which are jasmonic acid-responsive genes, were transiently expressed immediately after grafting, and the concentration of the bioactive jasmonic acid form, jasmonic acid-Ile, increased in hypocotyls 1 h after grafting [110]. *DAD1* (*DEFECTIVE IN ANTHER DEHISCENCE 1*) regulates JA biosynthesis and is suppressed by *ARF6* and *ARF8*, reflecting the influence of auxin concentration on grafting [111] (Figure 2). It has been shown that *SlWOX4* (*WUSCHEL-RELATED HOMEODOMAIN 4*, *WOX4*) is essential for vascular reconnection during grafting and may function as an early indicator of graft failure [112]. WOX4 acts redundantly with WOX14 in the regulation of vascular cell division [113]. Expression of *ATHB7* (*ARABIDOPSIS THALIANA HOMEOBOX 7*) and *ATHB8* are influenced by ABA during vascular system formation and are specifically expressed in the differentiating xylem. *PXY* (*PHLOEM INTERCALATED WITH XYLEM*) promoted cell division and organization in vascular meristems [107]. The *CLE41* (*CLV-3/ESR1-LIKE 41*)*/PXY* signaling module is activated by *MOL1* (*LRR-RLK MORE LATERAL GROWTH1*), which is required for cambium homeostasis [114]. *PXY* signaling, in turn, regulates *WOX14*, *TMO6* (*TARGET OF MONOPTEROS6*), and the *LBD4* (*LATERAL ORGAN BOUNDARIES DOMAIN4*) feedforward loop to control vascular proliferation [107] (Figure 2).

*VND6* (*VASCULAR-RELATED NAC-DOMAIN6*) acts downstream of *XVP* (*PRECOCIOUS XYLEM DIFFERENTIATION AND ALTERED VASCULAR PATTERNING*) during xylem differentiation [115]. XVP is a key regulator of vascular development, which modulates TDIF (TRACHEARY ELEMENT DIFFERENTIATION INHIBITORY FACTOR)–PXY signaling outputs and acts through binding to the PXY co-receptor BAK1 (BRI1-associated Kinase 1) [113,115]. The proteins CLE41 and CLE44 interact collaboratively with the TDIF receptor (TDR, also referred to as PXY), a process integral to the promotion of cambial cell proliferation. [103] (Figure 2). TDR interacts with GSK3s (glycogen synthase kinase 3 proteins) at the plasma membrane and activates GSK3s in a TDIF-dependent fashion. TOR (target of rapamycin) plays an important role in promoting vascular reconnection and cucumber/pumpkin graft union formation [86], whereas genes such as *APL3*, *STP1*, *DIN6,* and *SWEEET*, the latter being involved in sugar sensing, are repressed and involved in grafting success.

In a comprehensive overview, many genes are regulated or regulate other genes and substances during grafting, but relatively few can be used as indicator genes, as has been reported for *WOX4* as an early indicator of graft failure. Currently, most of the studied genes are involved with auxin, ethylene, and cytokinin, a finding which may be related to the important role of these hormones in grafting (Figure 2). However, more detailed and in-depth studies are still needed.

### 4.4. The Role of lncRNAs in Grafting

In contrast with coding genes, which generate mRNAs that can be translated into proteins, non-coding RNAs (ncRNAs) that cannot be translated into proteins exist in living organisms [116]. Long non-coding RNAs (lncRNAs) are RNA fragments with a length greater than 200 bp that do not have an open reading frame (ORF) and which have almost no ability to encode proteins [117]. lncRNAs, initially considered the “noise” of gene transcription, are the products of RNA polymerase II transcription that have either no biological function [118] or a function which is unclear [119]. Research has found that some specific lncRNAs can regulate the expression of upstream or downstream target genes, or act as a regulatory factor, being expressed in specific tissues, cells, and developmental stages, and participating in the regulation of specific plant life processes [117,120,121].

Previous research has shown that 22 mobile lncRNAs (mob-lncRNAs) were identified from the phloem tissue of grafted cucumber, which systematically move to the root tips and developing leaves in response to early phosphate deficiency in the plant [122]. A total of nine mob-lncRNAs were identified in the grafted tomato/potato system. Six mob-lncRNAs moved from the scion (tomato) to the rootstock (potato), and functional enrichment indicated that the mob-lncRNAs might be involved in vesicle transport, mitosis, and enzyme activity in potato. Three of the nine mob-lncRNAs moved upward from the rootstock to the scion in grafted plants, and their predicted function suggested that they might be involved in protein redox reactions, cell activities, and biological processes [123]. In addition, lncRNAs transfer between different cell types via exosomes as a means of information exchange, as important activators or inhibitors in regulating gene expression, and as participants in various biological processes [124,125].

As mentioned above, mob-lncRNAs play an important role in regulating various plant life processes. Mob-lncRNAs in grafted plants have also attracted increasing research attention, and mob-lncRNAs are also involved in the regulation of life processes in target tissues. The functional mechanism of mob-lncRNAs will become a new hotspot in research into substance movement within grafted plants.

### 4.5. Unveiling Grafting Mechanisms through Genomic Analysis

Transcriptome dynamics at *Arabidopsis* graft junctions reveal an inter-tissue recognition mechanism that activates vascular regeneration. Tissues above and below the graft junction rapidly develop an asymmetry such that many genes are more highly expressed on one side of the junction than on the other. This asymmetry correlated with the expression of sugar-responsive genes, and a recognition mechanism was activated independently of functional vascular connections [126]. Studies of enzymes associated with sugar metabolism by comparative transcriptomic analyses of graft combinations have indicated that a substance consisting of β-1,4-glucanases secreted into the extracellular region facilitates cell-wall reconstruction near the graft interface [85].

More and more studies have found that grafting can induce phenotypic variations in plants, which occur not only in grafts (mainly in scions) but can also be transmitted to offspring through asexual and sexual pathways [33]. It was found that grafting with red cabbage resulted in a variety of phenotypic changes, including shallow leaf-margin cleavage, thickened cuticle, increased number of branches, and advanced development [127]. Whole-genome DNA methylation levels were analyzed, and it was found that variation in DNA methylation of the coding genes associated with leaf-margin variation (*ARF10*, *ROF1* and *TPR2*) could be maintained up to the fifth generation of the sexual offspring. This study provides a new perspective on the formation and maintenance of graft-induced phenotypic variations and provides a theoretical basis (DNA methylation) for the fixation and exploitation of favorable graft-induced variations in vegetables [127]. Meanwhile, DNA methylation is crucially important for vasculature and meristem development, which is an important component process in graft healing [128].

## 5. Conclusions and Prospects

As one of the most economical and effective means by which to increase yield, improve quality, and enhance the stress resistance/tolerance of horticultural crop plants, the range of applications of grafting technology is still expanding. Since 1980, micropropagation and micrografting have been developed on the basis of tissue culture technology, which can be used in sterile tissue culture with 0.1 to 0.14 mm long in vitro stem-tip micrografting, which has been developed in China, the United States, and Spain. In addition to the development of micrografting, grafting mechanization (the application of mechanical equipment to achieve the automatic grafting of vegetables) will further increase productivity and free workers from the tedious work of grafting. Horticultural crop grafting will lead the whole vegetable planting industry into the mechanization of crop planting. With further developments, grafting mechanization will continue to be upgraded within the entire horticultural crop industry chain, and with the success of selecting rootstock scions from the seedling stubble to the grafting method to achieve industrial knowledge and digitalization, agricultural workers would only need to choose what kind of horticultural crop products are needed, while other intelligent digital processes can be selected to match.

In addition to the continuous improvement in grafting technology, further exploration of the nature of grafting affinity and the mechanism of rootstock influence should be the focus of future research. Grafting plays an important role in vegetables, but the processes involved still need to be clarified. Among the topics which need elucidation, the mechanism of the interaction between the rootstock and scion, the interaction between grafted plants and the environment (temperature, light, water), the uptake and transport of nutrients by grafted plants, and the roles of non-coding RNA and epigenetic inheritance demand significant in-depth research. The key determinant in grafting still remains germplasm resources, and further research is necessary to identify exemplary materials conducive to the selection and breeding of superior grafted varieties.

## Figures and Tables

**Figure 1 plants-12-02822-f001:**
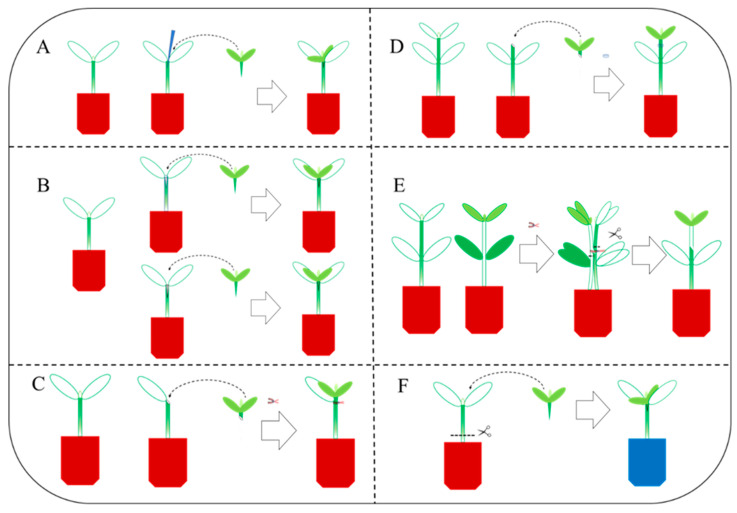
Diagrams of representative grafting methods. (**A**) Cuttage grafting. (**B**) Cleft grafting. (**C**) Patch grafting. (**D**) Casing grafting. (**E**) Approach grafting. (**F**) Double-root-cutting grafting.

**Figure 2 plants-12-02822-f002:**
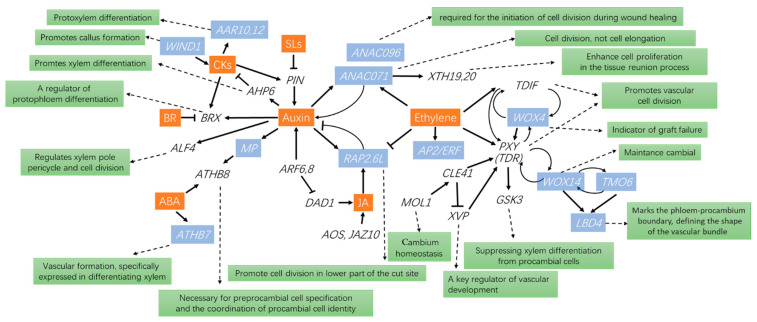
The interaction between hormones and genes involved in the grafting process. Hormones are shown in white on a red background. White letters on blue are transcription factor genes. Black italics are genes. Black letters on green indicate the function of the genes.

## Data Availability

Not applicable.

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
