# Peer review of "Study on the Applications and Regulatory Mechanisms of Grafting on Vegetables"

_plants, 2023, doi:10.3390/plants12152822_

Round 1
Reviewer 1 Report
Keywords: Vegetable; grafting; mechanism; regulation; applications
These are too general words, the reviewer recommends making them more specific, for example: grafting of vegetables,
Lines 28-32
Not a very clear phrase. Exactly the same goals can be achieved simply by vegetative propagation of valuable plant samples, without grafting.
Lines 127-131
Why is the text in italics?
Section 4. Mechanism of Grafting and Target Traits in Vegetables
The section partially (in a more expanded form) duplicates section 1 (The Importance and Uses of Grafting) especially in terms of stress tolerance.
I think the authors should rewrite section 1. Perhaps it is worth giving general information about plant grafting, not only vegetable crops, including interspecific grafting. Also, such important things as the mechanisms of communication of conducting systems during grafting, the creation of plasma between the tissues of the rootstock and scion, etc., should be covered here. In addition, if we talk about the use of the vaccination method, then this section does not say anything about using them for scientific purposes, for example, to resolve the issue of the systemic or local action of various signaling molecules. Please add this to the section.
Section 5.2, 5.3, 5.4.
It makes sense to break each of these sections into paragraphs, otherwise the information is difficult to perceive.
Section 5.5. The Role of Genomics in Grafting
The section should be renamed because it is not about genomics.
Reviewer 2 Report
Editor-in-Chief,
Plants
------------------------------------------------------
The manuscript entitled “Study on the applications and regulatory mechanisms of grafting on vegetables“ represents a valuable paper providing useful information on grafting that can overcome problems associated with unsuitable soil, increase tolerance and resistance of plants to biotic and abiotic stresses, improve the quality of vegetable products and increase the yield and value of vegetable crops. The method of inoculating vegetables against Fusarium wilt of watermelons was used in 1927 in Japan. The technique quickly spread to Japan and then to Korea and other East Asian countries to overcome the problems of intensive farming on limited arable land. In the late 20th century, vegetable grafting was introduced to Europe and North America. It enjoys growing interest, both from greenhouse growers and organic producers. Grafting vegetables is relatively easy as vegetables are mostly herbaceous. For economic reasons, grafting is best suited for growing in tall tunnels or greenhouses. Grafting applies to high-yielding seasonal vegetables such as tomato, watermelon, cucumber. The authors presented various methods of grafting vegetables, discussed the processes of graft survival and the factors that affect it. A valuable contribution of the authors is the discussion of the interaction of germplasm and rootstocks and the molecular regulation of vegetable grafting. I agree with the authors that grafting may be more use thanks to mechanization, which will allow greater economic benefits by improving the quality of products.
The paper is written correctly and includes all elements required in this type of scientific paper. Remarks for the authors are below. The paper in the submitted scope is interesting and is suitable for printing in “Plants” after the changes suggested by the Reviewer.
Remarks for the authors:
Title - suitable
Abstract - suitable
Keywords - suitable
Page 1, Line 17……..is…..Vegetable…. should be…vegetable
Chapters – suitable
Page 3, Line 103……. the parenthesis is closed but not opened
Page 4, Line 159…….is……the the……….should be…..the…
Page 6, Line 223…….is……should maintained……….should be….. should be maintained …
Page 7, Line 311……. is…..wilt…. should be…Fusarium wilt?
Page 8, Line 336……. is…..Aude…. should be…Habran et al.?
Page 8, Line 341……. is…..Soteriou…. should be…Soteriou et al.?
Page 9, Line 400……. is…..Rouphal et al.…. should be…Youssef et al.?
Page 9, Line 417……. is…..Schwartz.…. should be…Schwartz et al.
Page 10, Line 470……. is…..Arabidopsis…. should be…Arabidopsis
Page 10, Line 471……. is…..[89, 90]…. should be…[87]?
Page 10, Line 472…lack of full stop at the end of the sentence …is .. A…. should be… . A
Page 13, Line 584……. is…..Arabidopsis…. should be…Arabidopsis
Page 14, Line 621……. the parenthesis is opened but not closed
References – need improvement
All authors should be given
Page 15-19….Lines 651, 653, 657, 660, 662, 665, 667, 671, 675.... and so on…..
publication titles should be written in lowercase
Page 15, Line 655……. is…..Solanum habrochaites …. should be… Solanum habrochaites
Page 15, Line 665……. is…..Solanum melongena …. should be… Solanum melongena
Page 16, Line 687……. is…..Cucumis sativus…. should be… Cucumis sativus
Page 16, Line 700……. is…..PLANT GROWTH REGUL.…. should be… Plant Growth Regul.
Page 16, Line 710……. is…..Citrullus lanatus…. should be… Citrullus lanatus
Page 16, Line 719……. ..…..needs improvement - repeating
Page 16, Line 721……. is…..Artemisia scoparia…. should be… Artemisia scoparia
Page 16, Line 722……. is…..Chrysanthemum morifolium…. should be… Chrysanthemum morifolium
Page 17, Line 759……. is …. Cucumis melo L. subsp. melo var. indorus H. Jacq.…. should be… Cucumis melo L. subsp. melo var. indorus H. Jacq.….
Page 17, Line 765……. is …. Cucurbita spp...........Cucumis melo L. …. should be… Cucurbita spp........ Cucumis melo L.
Page 17, Line 788……. is …. Meloidogyne incognita ........... should be… Meloidogyne incognita
Page 17, Line 801……. is …. Capsicum annuum ........... should be… Capsicum annuum
Page 17, Line 803……. is…..Arabidopsis…. should be…Arabidopsis
Page 18, Line 805……. is….. Solanum lycopersicum …. should be… Solanum lycopersicum
Page 18, Line 837……. is….. Malus …. should be… Malus
Page 18, Line 841……. is….. Datura stramonium …. should be… Datura stramonium
Page 18, Line 849, 851, 853, 854, 861, ……. is…..Arabidopsis…. should be…Arabidopsis
Page 18, Line 863, ……. is…..Arabidopsis thaliana…. should be…Arabidopsis thaliana
Page 19, Line 865, 867,869, 875, 898, 904……. is…..Arabidopsis…. should be…Arabidopsis
Page 19, Line 882. is….. Phytophthora sojae …. should be… Phytophthora sojae
Page 19, Line 888, ……. is….. [Long non-coding RNAs in plants]…. should be… Long non-coding RNAs in plants
Reviewer 3 Report
This manuscript provides a broader perspective on several aspects of vegetable grafting focusing on impacts on plant stress resistance, crop yield and quality, and molecular regulatory mechanism in grafting. Overall, the manuscript is good. However, it requires some major changes in the structure and clarity of the sentence and the flow of the content. Several sections do not have references and lack important information while reporting results.
Few comments below:
L36: spell out CRP
L45: ‘the number of roots on grafted plants was twice than that on cucumber seedlings’. This sentence better follows the next statement ‘more than 30%.........deep in the soil horizon’ and serves as the justification for that sentence.
L51: fruit yield and quality
L56: ‘grafting seedling growth is strong, growth process is fast’ can be replaced with better word choices such as ‘grafted plants have vigorous growth’
L57-59: ‘Yields of grafted……..salt stress conditions’ This sentence is not clear
L69: what about the cost of grafted tomato plants or grafted plants in general compared to nongrafted plants?
L79-82: ‘The cuttage..…….into a wedge’. This sentence is not clear.
L103: remove the bracket
L150-L167: Is there a reference?
L194-195: Add an example of appropriate seedling age of rootstock and scion. This publication mentions the one for watermelon grafting http://www.vegetablegrafting.org/wp/wp-content/uploads/2018/04/WatermelonMelonGrafting3-15-18.pdf
L198-203: The description under grafting technology is too general. Are there specific examples that show variable success with different grafting technologies/methods?
L209-242: Please include references as needed
L286: Please specify the rootstock for grafted watermelon
L293 and L295: Please specify the rootstock for grafted eggplant
L326-328: It’s helpful for readers to know how fruit yield and quality differed with different grafting combinations.
L342: grafting onto which rootstocks?
L359-361: Employing melon as the rootstock for grafted pumpkins? Or the other way? And L363-365 ‘Melon rootstocks……post-grafting’ better flows after describing melon and pumpkin grafting combinations.
L363: https://www.mdpi.com/2311-7524/8/10/888 This paper shows grafting compatibility of melons with interspecific hybrid squash and similar or improved fruit quality for grafted plants. This will be a good addition in this description.
L368: ‘flesh firmness’ is a common terminology than ‘flesh hardness’
L404: Need more than one reference here as you mentioned ‘numerous studies’. https://www.mdpi.com/2311-7524/8/10/888 Again, this paper shows enlarged melon fruit and increased melon production by grafting with interspecific hybrid squash
L407-408: Better to provide name of scion and rootstock cultivars with strong/low affinity
L420: Does environment show variable effect on fruit yield and quality for similar scion/rootstock combinations?
L630: Please make sure of the last statement
It requires some major changes in the structure and clarity of the sentence and the flow of the content.
